# Pediatric Autoimmune Neuropsychiatric Disorders Associated with Streptococcal Infections (PANDAS): Myth or Reality? The State of the Art on a Controversial Disease

**DOI:** 10.3390/microorganisms11102549

**Published:** 2023-10-13

**Authors:** Saverio La Bella, Giovanna Scorrano, Marta Rinaldi, Armando Di Ludovico, Francesca Mainieri, Marina Attanasi, Alberto Spalice, Francesco Chiarelli, Luciana Breda

**Affiliations:** 1Department of Pediatrics, “G. D’Annunzio” University of Chieti-Pescara, 66100 Chieti, Italy; 2Department of Pediatrics, Buckinghamshire Healthcare NHS Trust, Aylesbury-Thames Valley Deanery, Aylesbury HP21 8AL, UK; 3Child Neurology Division, Department of Pediatrics, “Sapienza” University of Rome, 00185 Rome, Italy

**Keywords:** PANDAS, Pediatric Autoimmune Neuropsychiatric Disorders Associated with Streptococcal Infections, PANDAS syndrome, neuropsychiatric disorder, streptococci, streptococcal infection, molecular mimicry, CaM kinase, striatal cholinergic interneurons, obsessive–compulsive disorder

## Abstract

Pediatric Autoimmune Neuropsychiatric Disorders Associated with Streptococcal Infections (PANDAS) syndrome is one of the most controversial diseases in pediatric rheumatology. Despite first being described more than 25 years ago as the sudden and rapid onset of obsessive–compulsive disorder (OCD) and/or tic disorder symptoms as complications of a Group A beta-hemolytic Streptococcus (GAS) infection, precise epidemiological data are still lacking, and there are no strong recommendations for its treatment. Recent advances in the comprehension of PANDAS pathophysiology are largely attributable to animal model studies and the understanding of the roles of Ca++/calmodulin-dependent protein kinase (CaM kinase) II, disrupted dopamine release in the basal ganglia, and striatal cholinergic interneurons. The diagnosis of PANDAS should be made after an exclusion process and should include prepubescent children with a sudden onset of OCD and/or a tic disorder, with a relapsing/remitting disease course, a clear temporal association between GAS infection and onset or exacerbation of symptoms, and the association with other neurological abnormalities such as motoric hyperactivity and choreiform movements. Antibiotic medications are the primary therapeutic modality. Nonetheless, there is a paucity of randomized studies and validated data, resulting in a scarcity of solid recommendations.

## 1. Introduction

Pediatric Autoimmune Neuropsychiatric Disorders Associated with Streptococcal Infections (PANDAS) syndrome is a subject that generates significant controversy within the fields of pediatric rheumatology and neurology. In 1998, Swedo et al. published the first description of a cohort of children with PANDAS, defining the disease as the sudden and rapid onset of obsessive–compulsive disorder (OCD) and/or tic disorder symptoms as complications of a Group A beta-hemolytic Streptococcus (GAS) infection [1,2]. The existence of the disease, its pathophysiology, clinical manifestations, and classification criteria are frequent points of debate, and physicians and academics often have divergent opinions. The discussed pathogenesis of PANDAS is primarily supported by evidence of neurological involvement in patients with Sydenham’s Chorea (SC). Unlike PANDAS, SC is universally acknowledged as a severe complication of GAS infections. In fact, the American Heart Association’s 2015 revision to the Jones Criteria for Acute Rheumatic Fever still includes SC among the major classification criteria for both high-risk and low-to-moderate-risk populations [3]. SC typically develops not simultaneously with GAS infection but after a variable period of time, up to several months, spreading with involuntary movements of the limbs and trunk and frequently accompanied by muscle weakness and emotional instability [4]. Interestingly, a consistent subset of children with SC experience OCD and tics, pointing to a major point of connection with PANDAS as well as the aforementioned issues [5,6,7]. However, there are some major points that must be specified regarding this topic. The absence of consensus in the medical literature is widely attributable to the lack of large randomized studies, the high incidence of both neuropsychiatric disorders (primarily tics, OCD, and emotional lability) and GAS infections in children, the frequent inability to establish clear temporal and causal correlations between GAS infection and the onset of symptoms, the consistent proportion of asymptomatic GAS carriers among children, and the insufficient data regarding the real efficacy of antibiotics in children with PANDAS, particularly in comparison to placebos or other types of drugs. The purpose of this article is to provide a narrative review of the most recent knowledge on PANDAS syndrome’s pathophysiology, clinical features, and therapeutic approaches. In addition, this paper aims to critically analyze both the evidence and lack of support for this discussed disease.

## 2. Epidemiology

There is a lack of precise data regarding the epidemiology of PANDAS. The challenges associated with understanding and diagnosing this condition primarily stem from intricate clinical presentations and the difficulty in establishing causal relationships. Undoubtedly, the prevalence of both GAS infections and OCD/tic disorders in children is noteworthy. While it is widely acknowledged that PANDAS is a condition of low occurrence, the precise prevalence and incidence rates remain uncertain [8,9,10,11]. According to the first description, the prevalence of the disease appears to be higher in males, with a male-to-female ratio estimated to be between 2.6 and 4.7 to 1. The mean age of symptom onset for children diagnosed with PANDAS and exhibiting tics is approximately 6.3 years, whereas those presenting with OCD tend to experience symptom onset at around 7.4 years of age [1]. Additionally, more recent research confirmed a male predominance [12].

## 3. Pathophysiology

Currently, the pathophysiology of PANDAS is still a matter of debate. Some authors advocate for PANDAS belonging to the complex Pediatric Acute-Onset Neuropsychiatric Syndrome (PANS) spectrum, assuming that not only GAS may induce neuropsychiatric symptoms in children. However, this point is also debatable, and different classification criteria have been provided for PANS patients, making this aspect even more complex [13]. According to the latest accredited evidence, after a GAS infection, antibodies produced against GAS epitopes presumably cross-react against self-tissue, such as proteins expressed in the neurons of basal ganglia, through a molecular mimicry mechanism, suggesting an autoimmune pathogenesis similar to SC [14]. Several functional studies and experiments have tried to explain the immune pathophysiology of the disease, focusing on distinct neuronal targets potentially involved in the complex interaction between Streptococcus and the brain, building a model of an autoantibody effect on the central nervous system. They include lysoganglioside, tubulin, dopamine 1 and 2 receptors (D1R, D2R) [15,16,17,18].

### 3.1. Animal Model Studies

Over the years, animal model studies have described how GAS infections (such as pharyngitis, skin, and/or soft tissue infections) can stimulate an immune response in mice, with the production of antibodies directed to brain tissue and the consequent development of neurological disorders. Specifically, the serum of mice previously immunized with GAS contained autoantibodies that reacted against cerebellar tissue. Contextually, mice presented movement and/or behavior disorders [15,19]. Furthermore, mice exposed to GAS antigens showed impaired food handling. Clinical features in mouse models have been found to be related to brain–blood barrier (BBB) disruption with consequent autoantibody diffusion. Interestingly, lipopolysaccharides broke the BBB in vitro, suggesting that a bacterial infection could lead to its disruption in vivo [20,21]. Moreover, functional studies revealed a new promising target of GAS infection, showing that intrathecal passive transfer of antibodies extracted from sera of children with PANDAS interfered with Ca++/calmodulin-dependent protein kinase (CaM kinase) II activity, leading to an increased protein level. CaMKII is a multifactorial protein mainly involved in behavior control, learning and memory mechanisms, neuronal development, cortical excitability, neurotransmitter synthesis, and release. Its activity is strictly related to calcium intracellular and extracellular levels, and an increased level of CAMKII has been associated with tics and/or OCD in mouse models. Specifically, mice treated with PANDAS serum showed increased CAMKII levels with a consequent increase in tyrosine hydroxylase activity and dopamine release. This disrupted signaling led to the onset of neuropsychiatric symptoms in animal models, while mice treated with control sera did not manifest any abnormal behavior or CAMKII/dopamine release impairment. Of note, antineuronal antibodies appeared to be cross-reactive with antibodies directed to a GAS carbohydrate antigen, N-acetyl-beta-D-glucosamine, which is presumably a crucial factor in GAS-mediated cellular signaling, with structure sequences similar to neuronal antigens [15,18]. CAMKII levels are considerably increased in SC while appearing to be slightly increased in PANDAS, suggesting that the different concentrations probably lead to neuropsychiatric and movement disorders in SC and isolated neuropsychiatric disorders in PANDAS. Indeed, CAMKII presents a high specificity of function related to qualitative and/or quantitative changes. Interestingly, several studies have suggested that the D2R represents the target of autoantibodies produced in PANDAS and SC, with subsequent development of choreiform movements and behavior disorders [16,19,20,21,22]. An experiment showed how the autoantibodies reacted with both intracellular and extracellular epitopes of D2R, and mice presented gait impairment with neuropsychiatric manifestations during in vivo tests. Specifically, the autoantibodies affected D2R neurons in the cortex and basal ganglia but did not manifest reactivity against D1R neurons [16]. This study differed from previous ones by excluding D1R from the targets implicated in PANDAS physiopathology. Even though several neuronal targets have been identified to be involved in GAS-related disorders, these findings have never been consistently addressed.

The latest evidence arising from the literature has allowed for a growing comprehension of PANDAS pathogenesis. Nonetheless, as stated above, the first study of a PANDAS cohort was published more than 25 years ago, and the authors provided a good point of view in the discussion: “studies of SC revealed that obsessive-compulsive symptoms were common during the illness (present in nearly three-quarters of children) and had their onset shortly before the chorea began […]. Based on these observations, we postulated that children might exhibit only tics or OCD if the “dose” of a presumed etiologic agent was not sufficient to cause frank chorea.” [1].

### 3.2. Striatal Cholinergic Interneurons: Latest Evidence of a New Key Player in PANDAS Pathophysiology

Presumably, the impairment of dopaminergic circuits in the basal ganglia related to autoantibodies could provoke the onset of extrapyramidal movement and/or neuropsychiatric symptoms, as occurs in other neurological diseases with a known pathogenesis. The latest evidence shows a misidentified neuronal population plays a key role in PANDAS pathophysiology [23]. Specifically, the IgG of children with PANDAS, compared to age- and sex-matched controls, were complexed with mouse cerebral slices, evaluating both in vivo and ex vivo bonding to substrates. Ribosomal S6 (RS6) protein was used as a validated marker of neuronal activity [24]. Unlike in previous studies, experiments were also carried out in human brains, proving how 11 PANDAS sera showed high-avidity binding to a newly identified neuronal population, striatal cholinergic interneurons (CINs). Over the years, functional studies have suggested that the bond between autoantibodies in children with PANDAS and CINs caused CIN activity inhibition. Notably, CIN loss of function has previously been found to be involved in Tourette syndrome pathogenesis and in repetitive behavioral disorders [25,26]. Experimental depletion of CINs in mice with normal neurodevelopment led to elevated grooming and repetitive behaviors. Furthermore, striatum postmortem analysis revealed a decreased expression of CIN-related genes in patients with Tourette syndrome. Interestingly, the IgG of children with PANDAS did not bind to either GABAergic interneurons or medium spiny neurons expressing D1R-D2R, unlike what was previously reported. Thus, the expression of D2R by CINs only could explain the high-avidity bond between IgG and CINs, sustaining the relevant role of DR2 in PANDAS pathophysiology compared with DR1. Interestingly, the same striatal mouse slices were complexed with PANDAS and control sera after treatment with immunoglobulins (IVIg) ex vivo, using RS6 activity as a marker. A relevant decrease in the CIN-PANDAS IgG bond occurred, with a contextual clinical improvement. This suggests an immune pathogenesis with a pivotal role for neuroinflammation in the onset of GAS-related disorders.

### 3.3. Imaging Studies

Imaging studies have been performed in children with PANDAS and revealed striatal involvement [27]. Specifically, a PET scan was carried out to evaluate neuroinflammatory changes in the basal ganglia and thalamus in children with clinically diagnosed PANDAS. Using a target expressed in activated microglia, it was observed that the PANDAS group had more neuroinflammation in both the caudate and lentiform nuclei on both sides of the brain. Of note, after IVIg treatment, a relevant decrease in striatal neuroinflammation occurred. Furthermore, a neuroradiological study compared the brain magnetic resonance imaging of 34 children with PANDAS to that of 82 age- and sex-matched controls [28]. The sizes of the caudate, putamen, and globus pallidus, but not the thalamus or total cerebrum, were considerably greater in the group of children with PANDAS than in the healthy controls. These findings are consistent with what was previously found in subjects with SC compared with normal subjects, suggesting an antibody-related inflammatory involvement of basal ganglia after GAS infection with subsequent development of typical symptoms. Therefore, these imaging studies confirmed the immune pathogenesis and the crucial role of basal ganglia neuroinflammation in children with PANDAS, as described in the latest clinical and functional studies.

### 3.4. The Complex Network of Basal Ganglia: From Physiology to Pathophysiology

Several theories have been proposed to explain how neuroinflammation of the basal ganglia could lead to the onset of neuropsychiatric symptoms in children with PANDAS. Basal ganglia are subcortical nuclei deeply located in the cerebral hemispheres and include the putamen, caudate, globus pallidus, and related structures such as the subthalamic nucleus (SN) in the diencephalon, the substantia nigra pars compacta (SNc), and the pedunculopontine nucleus in the mesencephalon [29]. Basal ganglia modulate motor learning and control, executive functions, behavior, and emotions. They include output, input, and intermediate nuclei. The input nuclei (striatum, nucleus accumbens, and olfactory tubercle) receive inputs from the cortex, thalamus, and SNc, while the output nuclei present as outputs in thalamocortical circuits and include the internal parts of the globus pallidus and substantia nigra pars reticulata (SNr). Intermediate nuclei comprise SN and SNc. The integrity of basal ganglia circuits requires a physiological release of dopamine. Specifically, the striatum plays a key role in basal ganglia circuits, representing the great subcortical structure in the mammalian brain. It includes two types of neurons: projection neurons, also called medium-sized spiny neurons (MSNs, 90%), and interneurons (10%).

MSNs are GABAergic inhibitory neurons and include two distinct groups. One MSN group projects to the globus pallidus externus (GPe) through D2R, leading to the tonic inhibition of the globus pallidus internus (GPi) and SNr (indirect pathway), while the second MSN group projects to GPi and SNr through D1R, suspending the tonic inhibition of GPi and SNr on their targets. This circuit constitutes the direct pathway that promotes elected motor programs (Figure 1) [30,31]. The most abundant interneurons are cholinergic. They use acetylcholine (Ach) as their main neurotransmitter and are tonically active, modulating both MSNs’ and other GABAergic interneurons’ activity.

Cholinergic interneurons present an associative function, integrating synaptic inputs, and their axon projections mainly target MSNs [24,32,33].

Other basal ganglia functions include attention and time estimation, implicit learning and habit formation, reward-related behavior, and emotions. Indeed, they relate to cortical and subcortical areas through several loops that constitute a complex network [29]. Cholinergic interneurons regulate the activity of MSNs and their D1 and D2 expression, both directly and through interactions with glutamate and dopamine release, and modulate GABAergic interneurons [30]. Furthermore, nicotinic Ach receptors lead to dopamine release, whereas muscarinic receptors inhibit it at the presynaptic level [30,34]. Interestingly, cholinergic disruption in the dorsal striatum may thus produce an imbalance between the direct and indirect pathways through the basal ganglia, leading to disinhibition of off-target behaviors. Specifically, tics, obsessions, and compulsions have been associated with hyperactivity of the direct pathway concerning motor and limbic domains. In mouse models, cholinergic transmission interrupted stereotypies, and the administration of the D2 antagonist raclopride led to an increase in Ach, with the same result. By contrast, local blockade of muscarinic receptors or neurotoxin-induced degeneration of cholinergic interneurons worsened the motor stereotypes [35]. These results suggest a critical role for cholinergic mechanisms in the pathophysiology of movement and behavioral disorders that affect the basal ganglia network. The identification of a novel neuronal population (CINs) involved in GAS-related disorders sheds light on a greater clarification of PANDAS pathophysiology with prognostic and therapeutic implications (Figure 2). Moreover, validated molecular targets could help highlight new diagnostic and therapeutic biomarkers. However, the specificity of the CIN-IgG bond and the neuroinflammation pattern should be elucidated.

## 4. Clinical Features and Classification Criteria

The working PANDAS classification criteria require all of the following [1]:(1)Presence of OCD and/or a tic disorder;(2)Prepubertal symptom onset;(3)Acute onset of symptoms with a relapsing/remitting disease course;(4)A clear temporal association between GAS infection and symptom onset or exacerbation;(5)Association with other neurological abnormalities (particularly motoric hyperactivity and choreiform movements).

The diagnosis of PANDAS is based only on clinical criteria, requiring a critical exclusionary diagnostic process to rule out other psychiatric and neurological disorders. Several studies have attempted to establish the differences between PANDAS and other psychiatric disorders. However, the similarities between OCD and tics in PANDAS and non-PANDAS disorders are notable. The lack of identifiable biological markers or distinct clinical characteristics continues to raise doubts and provoke controversy over the accuracy of diagnoses [9,14,36,37,38]. Working criteria for PANDAS were improved in 2017: OCD and/or tics should be complex or not observable in other disorders; a specific age should be considered (symptoms of PANDAS are more common between the age of 3 years and puberty); acute onset and episodic changes in behavior should be both present; the association with GAS infection should be clearly proved; and neurological impairment should be present [39,40].

### 4.1. The Causal and Temporal Relationship between PANDAS and GAS Infection

The relationship between GAS infections and PANDAS or other neuropsychiatric manifestations is still debatable. A large case–control study was conducted in 2005 on about 75,000 children aged 4 to 13 years old. Individuals diagnosed with OCD, Tourette syndrome, and tic disorder exhibited a higher probability of having experienced a previous GAS infection within the three months preceding the onset of symptoms in comparison to the control group, while children who experienced multiple streptococcal infections within a 12-month period had a significantly higher risk. The observed associations remained largely consistent even when the analysis was restricted to cases with a clearly defined date of symptom onset or when the matching criteria for health care behavior were more stringent. The authors conclude that their findings lend epidemiologic evidence that PANDAS may arise as a result of a postinfectious autoimmune phenomenon induced by childhood streptococcal infection [41].

In a large prospective study of 814 patients with a sore throat aiming to determine the characteristics of PANDAS patients, children who were diagnosed with a GAS infection and subsequently received treatment for their infection did not exhibit a higher likelihood of developing PANDAS in comparison to children who were presumed to have a viral illness or were in good health [8].

Another longitudinal study, which was published in 2007, involved a sample of 693 children. The study’s findings indicated that there were observable motor and behavioral changes that occurred in association with a positive GAS culture. Moreover, the study provided support for the notion that repeated GAS exposure increases the risk of these changes [42].

In 2017, a population-based cohort study focusing on the risk of mental disorders was performed using data from the nationwide Danish registers, including more than 1 million children [43]. Individuals who were diagnosed with streptococcal infection demonstrated an increased vulnerability to a range of mental disorders, particularly OCD, to a greater extent, and tic disorders, to a lower extent, in comparison to individuals who did not undergo testing for streptococcal infection. Moreover, the likelihood of developing any neuropsychiatric disorder was found to be higher following a streptococcal throat infection compared to a nonstreptococcal infection. However, it is worth noting that individuals who have a nonstreptococcal throat infection also exhibit a heightened susceptibility to developing any form of mental disorder [43].

However, additional studies have reached divergent results, making the relationship between GAS infection and neuropsychiatric disorders, including PANDAS, still debatable [8,44,45,46].

There exist three primary approaches for ascertaining the presence of a GAS infection. In general, the identification of a concurrent throat infection, such as pharyngitis or pharyngotonsillitis, is accomplished through the utilization of a throat swab. Both a rapid antigen detection test (RADT) and a culture test can be utilized in this context. The RADT offers advantages in terms of cost-effectiveness and efficiency, whereas the culture test is considered to be more sensitive but also entails higher expenses and longer processing times. An inaccurate acquisition of the sample may lead to false-negative results, especially in young children, such as infants and toddlers, who may not willingly comply with the invasive nature of the testing procedure [47,48,49]. However, GAS may cause only paucisymptomatic throat infections (without the execution of a throat swab) or, in addition to the pharynx and tonsils, may affect the skin, nasal cavity, perineum, and genitals [50]. GAS may cause infections in these regions, causing scarlatiniform rash, impetigo, deep tissue infiltration, and perianal or genital dermatitis [51]. Recent GAS infections may be detected with a blood test investigating increased anti-streptolysin-O (ASO) or anti-DNase B (ADB) titers [47,48]. It may be challenging to correctly interpret ASO and ADB titers. Indeed, the onset and duration of these biomarkers are distinct. After a GAS infection, ASO typically begins to rise after 1 week and peaks 3 to 5 weeks later, decreasing after 6 to 8 weeks and staying elevated for about 6 months or more [52].

ADB usually rises slightly later than ASO, about 2 weeks after GAS infection, peaks 6 to 8 weeks later, and begins to decline about 3 months after the infection (Table 1).

Thus, a concomitant or recent GAS infection must be documented before making a diagnosis of PANDAS.

There are some additional considerations that should be made:(1)GAS infection is common among children attending communities and schools.(2)Similarly, vocal and/or motor tics and OCD are not uncommon in children.(3)ASO and ADB may be found to be elevated in older GAS infections as well, resulting in false-positive tests.(4)ASO may rise due to chronic liver disease, hypergammaglobulinemia, and hypercholesterolemia [52].

In patients with a suspicion of PANDAS, performing repeated tests (throat swabs, ASO, and ADB) appears to be the best way to evaluate the relationship between clinical manifestations and GAS infections. Indeed, flares ought to be correlated with a positive throat swab or an increase in ASO/ADB titers relative to previous wellness analysis [53]. Nevertheless, discerning the correlation between a disease flare and a GAS infection can prove challenging, primarily due to the lack of swab or blood sample procedures in routine clinical practice. This limitation may arise from the suboptimal adherence by patients and families.

### 4.2. Clinical Characteristics

PANDAS predominantly manifests during a specific developmental stage in childhood, being typically observed between the age of 3 years and the onset of puberty. Moreover, the incidence of PANDAS is notably higher among males [39,54]. The occurrence of PANDAS at an early age is linked to the period when the rate of exposure to GAS is at its peak [39]. Children impacted by PANDAS exhibit notable and expeditious alterations in their behavior, showing psychotic symptoms within a timeframe ranging from 24 to 72 h [55]. Following this, the clinical trajectory can be described as exhibiting a “sawtooth” pattern, wherein periods of symptom inactivity are interspersed with episodes of exacerbation that manifest suddenly and resolve gradually over a span of weeks to months [1,14,56].

The symptoms commonly associated with PANDAS encompass a sudden and pronounced emergence of OCD, accompanied by tics, hyperactivity, urinary urgency, impulsivity, anxiety, impulsiveness, eating disorders, and a notable deterioration in academic performance, as well as a decline in handwriting proficiency [57,58]. An extended duration of a streptococcal infection may exacerbate the clinical outcome [39]. On average, the severity of anxiety and other mood symptoms tends to be higher than that of somatic and functional symptoms, such as gastric pain and arthromyalgia [12]. Psychiatric comorbidities observed within PANDAS cohorts encompass attention-deficit/hyperactivity disorder (ADHD) at a prevalence rate of 40%, oppositional defiant disorder (ODD) also at a prevalence rate of 40%, and depression at a prevalence rate of 36%. According to the literature, additional symptoms may include emotional labiality at a rate of 66%, personality changes at 54%, bedtime fears at 50%, fidgetiness at 50%, sensory defensiveness at 40%, irritability at 40%, and impulsivity and distraction at 38%. The prevalence of anorexia and dietary limitations was comparatively lower, affecting approximately 10–20% of individuals. However, these conditions frequently resulted in substantial weight reduction and posed significant health hazards [1,59]. Enuresis, occurring in 20% of cases, and dysthymia, observed in 12% of cases, are less frequently encountered, and urinary urgency and dysuria have also been reported [1,59,60]. It is not uncommon for individuals to exhibit behavioral changes resembling those associated with ADHD as a result of concurrent or previous infections caused by GAS [8]. Up to 95% of patients diagnosed with PANDAS exhibit choreiform movements bearing resemblance to those observed in SC but are not present at rest and are always elicited by stressed postures [1,59]. However, SC is characterized by more severe obsessive–compulsive syndromes and the presence of hypotonia. Additionally, SC typically exhibits complete remissions and a duration of less than one year. On the other hand, the course of PANDAS is more likely to be chronic, with an average remission period of 3.3 years [4,61,62]. Furthermore, it was noted that a majority of children, approximately 72%, exhibited at least one instance of worsened PANDAS symptoms during the course of gradual remission in a period of 6–57 months of follow-up [63].

Numerous studies have tried to ascertain the existence of distinct clinical characteristics in PANDAS in contrast to OCD or tics without a PANDAS association. Nevertheless, there have been no significant clinical differentiations identified between these two cohorts, with the exception of the observation that children diagnosed with PANDAS commonly manifest psychiatric conditions beyond tic disorder or OCD [44,64,65]. Research conducted by Bernstein et al. in 2010 confirmed that children diagnosed with PANDAS demonstrated an increased probability of manifesting symptoms including separation anxiety, urinary urgency, hyperactivity, impulsivity, deterioration in handwriting, and decline in school achievement during the initial phases of the illness. This was in comparison to children diagnosed with non-PANDAS OCD. Moreover, in a study conducted by Murphy et al., it was shown that children diagnosed with Pediatric Autoimmune PANDAS demonstrated a higher frequency and intensity of tic symptoms in comparison to other individuals [57,64].

Furthermore, the caregivers of patients with PANDAS encounter heightened levels of anxiety, stress, and depression [66].

The aforementioned studies did not yield conclusive evidence supporting the presence of distinct phenomenological features specific to PANDAS. Due to the challenges associated with diagnosis, it is evident that a more refined definition of PANDAS syndrome is necessary to establish more accurate diagnostic guidelines and indications.

## 5. Treatment Options and Disease Course

Antibiotic medications serve as the primary therapeutic modality for pediatric patients diagnosed with PANDAS. Nevertheless, it is important to emphasize that there is a dearth of randomized studies and validated data for such medications in comparison to placebos, resulting in a lack of robust recommendations. Furthermore, there are currently no established guidelines regarding the right choice of antibiotics for administration. In clinical practice, the selection of an antibiotic is primarily based on individual experience, patient and family preferences, and limited non-randomized studies.

According to the literature, beta-lactam antibiotics (mainly cephalosporins), azithromycin, and clindamycin may all be useful for affected patients [60]. However, some experts discourage the use of amoxicillin in PANDAS because it does not exhibit enhanced intracellular permeation and only demonstrates its activity during bacterial cell division.

As stated above, there have only been a few randomized trials evaluating the different treatment strategies in children with PANDAS [67].

The first study, conducted in 1999, employed a double-blind, balanced cross-over design and involved a sample of 37 patients diagnosed with PANDAS. Although there was a modest effectiveness observed for antibiotic medications, this study did not provide statistical evidence of improvements in children receiving penicillin V prophylaxis compared to those receiving a placebo following an 8-month follow-up period. It is worth noting that there was an equivalent incidence of infections observed in both the treatment and placebo groups, and no significant alterations were observed in the severity of OCD or tic symptoms [68].

A subsequent study conducted in 2005 demonstrated notable advances in the reduction in both GAS infections and neuropsychiatric symptoms among patients who received antibiotic prophylaxis using penicillin or azithromycin, as observed during a one-year follow-up period. However, a comparison between the treated group and a placebo group was not conducted, and the baseline period was retrospectively evaluated [69].

Patients suffering from PANDAS usually show a dramatic resolution of their symptoms after antibiotic administration despite the frequent relapsing of a consistent group of them. Indeed, according to a small observational study on 12 PANDAS patients, about 50% of them may have one or more recurrence, triggered by a GAS infection, in a follow-up period of 3 years [60]. The same authors, in a following study, observed that children with PANDAS exhibited a statistically significant higher likelihood of experiencing a remission of neuropsychiatric symptoms when undergoing antibiotic therapy as compared to individuals without PANDAS [64].

Although there is a lack of high-quality data, it is plausible that the risk of recurrence is associated with the frequency of GAS infections prior to the onset of neuropsychiatric symptoms [60].

According to a larger number of authors, the presence of untreated GAS infection has been observed to potentially induce OCD and tic disorders within a specific subset of the pediatric population [8,67,70]. Moreover, additional research has documented the favorable effectiveness of antibiotics in individuals diagnosed with PANDAS or those exhibiting symptoms suggestive of the condition. These findings have also indicated a high level of patient satisfaction with the antibiotic treatment [71].

In another prospective study including a cohort of 120 children diagnosed with PANDAS, a comparison was made between 56 individuals who underwent tonsillectomy and/or adenoidectomy and 64 unoperated controls. Over a period of more than two years, the group that underwent surgery did not exhibit any significant differences compared to the control group in terms of the progression of symptoms, levels of streptococcal and neuronal antibodies, or the severity of neuropsychiatric symptoms. According to the authors, the criteria for performing tonsillectomy and/or adenoidectomy should be restricted to those that are endorsed for the broader community, including conditions like sleep-disordered breathing or recurrent GAS infections [72].

Psychological, behavioral, and psychopharmacological therapies that are customized to suit the specific characteristics of each child can lead to a reduction in symptoms and an enhancement in overall functioning, both in the short-term and long-term phases of the illness. Typically, evidence-based therapies are suitable for addressing the many symptoms associated with PANDAS. The presence of individual variations in the anticipated reaction to psychiatric medicine may necessitate a significant decrease in the initial dosage of treatment [40].

According to a considerable number of placebo-controlled studies, immunomodulatory therapy (IVIg and plasmapheresis) may be helpful in PANDAS and in children with infection-triggered OCD and tic disorders [73,74,75,76,77]. Some case reports showed full remission after treatment with IVIg, even with often ongoing corticosteroid therapy [74,78]. In 1999, Perlmutter et al. conducted a study employing a double-blind, placebo-controlled design to examine the efficacy of IVIg and plasmapheresis in mitigating OCD symptoms among patients diagnosed with PANDAS. The results indicated that both IVIG and plasmapheresis are effective, with OCD reductions of 45% and 58%, respectively [73]. Conversely, the administration of a placebo infusion did not yield any discernible impact. The trial’s outcomes exhibited substantial strength, leading to the American Society of Apheresis’ decision to designate plasmapheresis as a Category I, primary therapy choice for PANDAS [79]. However, the results are still controversial, and other studies failed to demonstrate considerable improvement compared to a placebo or presented low-quality standards [80]. Broader studies should be performed to clarify this aspect.

Corticosteroids, rituximab, and mycophenolate mofetil have also been evaluated for a possible role in the treatment of PANDAS [74]. Specifically, corticosteroids are suggested to be used early in the initial presentation (less than 2 weeks) in moderate-to-severe or extreme flares and even in mild flares if the clinical course lasts more than 2 weeks [74].

Nevertheless, ongoing clinical trials are currently examining the potential impact of rituximab on suspected autoimmune subtypes of OCD, aiming to determine whether this form of immunotherapy can contribute to a more favorable outcome [13].

However, given the significant implications of immunologic factors in the pathogenesis of PANDAS, there has been a growing interest among allergists and immunologists in exploring this disease [81].

A systematic review and meta-analysis of the treatment options for PANDAS patients was recently published in 2022 [82]. In contrast to previous observations, the authors’ analysis suggests that surgical intervention in specific patient populations has the potential to yield favorable outcomes [72,82,83]. However, the efficacy of medical therapy is still a subject of debate in academic circles, primarily due to the absence of universally accepted treatment guidelines and the variability in patient response based on the specific medication employed and the timing of its administration [82].

Furthermore, the implementation of a short-term trial of prophylactic antibiotics can contribute to the diagnostic process, which involves assessing the presence of reduced symptomatology following their administration.

## 6. Conclusions

Despite significant advances in the understanding of the pathogenesis and therapeutic management of affected children, PANDAS remains a subject of dispute among experts. The controversial issues discussed in this paper can primarily be attributed to the challenges associated with establishing a definitive and timely correlation between the occurrence or recurrence of OCD and tic disorder in children who experience multiple GAS infections. Furthermore, it is important to note that the existing research examining the effectiveness of treatment techniques suffers from limitations in terms of sample size and rigorous inclusion criteria, which consequently hinders the provision of robust recommendations. Additional research is required to enhance our comprehension of the real prevalence of the disease, the precise pathogenic pathways involved, and the appropriate evidence-based methodology for its treatment.

## Figures and Tables

**Figure 1 microorganisms-11-02549-f001:**
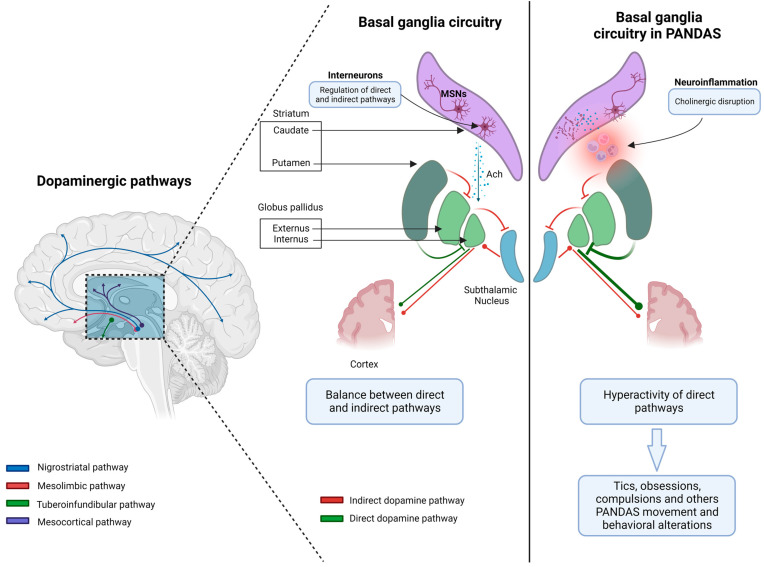
The pathophysiology of Pediatric Autoimmune Neuropsychiatric Disorders Associated with Streptococcal Infections (PANDAS) syndrome involves an autoimmune response triggered by a streptococcus infection. The presence of neuroinflammation in the dorsal striatum can result in the disruption of cholinergic activity. This disruption can lead to an imbalance between the direct and indirect pathways within the basal ganglia. Consequently, off-target behaviors such as tics, obsessions, and compulsions may be disinhibited. Notably, the hyperactivity of the direct pathway in relation to motor and limbic domains has been linked to these behaviors (created with BioRender.com). Abbreviations. Ach: acetylcholine, MSNs: medium-sized spiny neurons.

**Figure 2 microorganisms-11-02549-f002:**
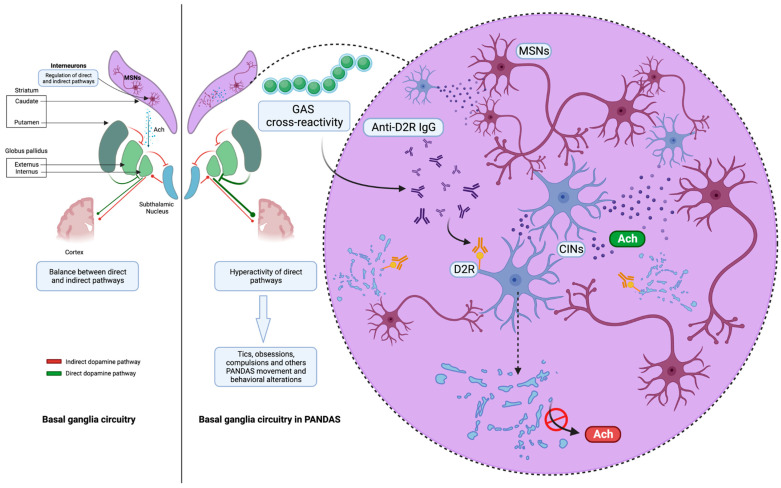
The IgG antibodies derived from pediatric patients diagnosed with Pediatric Autoimmune Neuropsychiatric Disorders Associated with Streptococcal Infections (PANDAS) exhibit affinity towards cholinergic interneurons (CINs) located within the striatum. The deficiency of CINs has been found to be independently linked to the presence of tics in people and the manifestation of repeated behavioral disease in mice. This suggests that CIN deficiency could potentially serve as a viable pathological factor (created with BioRender.com).

**Table 1 microorganisms-11-02549-t001:** Timing of anti-streptolysin-O (ASO) and anti-DNase B (ADB) titers, expressed in weeks.

Timing	Anti-Streptolisin O	Anti-DNAse B
Starting to increase (weeks)	1	2
Peak (weeks)	3–5	6–8
Starting to decrease (weeks)	6–8	12

## Data Availability

No data were generated during the realization of this paper.

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
