# Peer review of "Pediatric Autoimmune Neuropsychiatric Disorders Associated with Streptococcal Infections (PANDAS): Myth or Reality? The State of the Art on a Controversial Disease"

_microorganisms, 2023, doi:10.3390/microorganisms11102549_

Round 1
Reviewer 1 Report
Although there is much to be learned about PANDAS, this review was well written and comprehensive.
The scientific overview highlighted mouse studies primarily in Sydenham's chorea, but also included studies in PANDAS and the "dose" effect was of interest. The section on the "Complex Network of the Basal Ganglia" should be revised and focused. The relevancy in PANDAS made more clear rather than descriptive overview.
The tables would be better understood if the labeling was redone, clearer and better described.
Many are including Pediatric Autoimmune Neuropsychiatric Syndrome (PANS) in with PANDAS. Discussion of whether or not this is valid would be of interest.
In addition, there was mention of, but little review of the use of IVIg and steroids and these are often strongly advocated by parental groups to be treatments of choice.
At line 146, the word "replied" should be changed.
Author Response
Although there is much to be learned about PANDAS, this review was well written and comprehensive.
- The scientific overview highlighted mouse studies primarily in Sydenham's chorea, but also included studies in PANDAS and the "dose" effect was of interest. The section on the "Complex Network of the Basal Ganglia" should be revised and focused. The relevancy in PANDAS made more clear rather than descriptive overview.
Thank you for yout suggestion. We made the paragrapher more clear and focused.
- The tables would be better understood if the labeling was redone, clearer and better described.
Thank you for your comment. We have modified the Table 1 (previous Figure 3) as you suggested.
- Many are including Pediatric Autoimmune Neuropsychiatric Syndrome (PANS) in with PANDAS. Discussion of whether or not this is valid would be of interest.
Thank you for your comment. We have specified this point in the text (lines 78-83). The authors' opinion is that PANS and PANDAS could be different entities, and evidence for PANS is even weaker than that for PANDAS. Thus, we decided not to write about the argument in order to not give partial or incorrect opinions to the readers.
- In addition, there was mention of, but little review of the use of IVIg and steroids and these are often strongly advocated by parental groups to be treatments of choice.
Thank you for your suggestion. We have provided a wider description of immunomodulatory treatment for PANDAS syndrome.
- At line 146, the word "replied" should be changed.
Done
Reviewer 2 Report
Dear Dr Saverio La Bella,
Thank you so much for sending us your well-written paper. I believe that you and your colleagues discuss PANDAS in great detail. I recommend it to be published in its present form.
Best regard
Author Response
Thank you so much for your comments.
Reviewer 3 Report
The manuscript "Pediatric Autoimmune Neuropsychiatric Disorders Associated with Streptococcal Infections (PANDAS): myth or reality? The state of the art on a controversial disease" by S. La Bella et al. is the review on specific complication of group A beta-hemolytic streptococcal infections. The authors describe the possible molecular mechanisms involved into pathogenesis and possible therapeutic options. The work is looking well-illustrated and readable. It could be accepted for publication after revision.
1. The type of the review is not clear. Does it systematic, critical or other type? That's defining the point of view.
2. Structured reviews and meta-analyses should conform to the PRISMA guidelines, according the journal's instructions for authors. First of all, the choice of the literature and time frame should be disclosed.
3. Guess, the recent reviews on the topic should be mentioned in the introduction section. For example, https://doi.org/10.3892/etm.2020.9526
4. Minor corrections:
Affiliations: Please, delete the examples text ("Affiliation 1;" etc.);
Reference 10: Add the doi (10.3389/fped.2021.746639)
Author Response
The manuscript "Pediatric Autoimmune Neuropsychiatric Disorders Associated with Streptococcal Infections (PANDAS): myth or reality? The state of the art on a controversial disease" by S. La Bella et al. is the review on specific complication of group A beta-hemolytic streptococcal infections. The authors describe the possible molecular mechanisms involved into pathogenesis and possible therapeutic options. The work is looking well-illustrated and readable. It could be accepted for publication after revision.
- The type of the review is not clear. Does it systematic, critical or other type? That's defining the point of view. Structured reviews and meta-analyses should conform to the PRISMA guidelines, according the journal's instructions for authors. First of all, the choice of the literature and time frame should be disclosed.
Thank you for your comment. This is a narrative review of the literature, we have specified this point into the introduction (line 60).
- Guess, the recent reviews on the topic should be mentioned in the introduction section. For example, https://doi.org/10.3892/etm.2020.9526
Thank you for your suggestion. We have added this reference [2].
- Minor corrections:
Affiliations: Please, delete the examples text ("Affiliation 1;" etc.); Reference 10: Add the doi (10.3389/fped.2021.746639).
Done.
Round 2
Reviewer 3 Report
Repetitio est mater studiorum
The main problem with this work is that it does not meet the strict PRISMA guidelines. However, it may be useful in the above manner and could be accepted for publication in its current form.